# Mixed-Method Evaluation of a Community Pharmacy Antimicrobial Stewardship Intervention (PAMSI)

**DOI:** 10.3390/healthcare10071288

**Published:** 2022-07-12

**Authors:** Catherine V. Hayes, Donna M. Lecky, Fionna Pursey, Amy Thomas, Diane Ashiru-Oredope, Ayoub Saei, Tracey Thornley, Philip Howard, Aimi Dickinson, Clare Ingram, Rosalie Allison, Cliodna A. M. McNulty

**Affiliations:** 1HCAI & AMR Division, Primary Care and Interventions Unit, UK Health Security Agency, Gloucester GL1 1DQ, UK; donna.lecky@phe.gov.uk (D.M.L.); fionna.pursey@phe.gov.uk (F.P.); amy.thomas15@nhs.net (A.T.); diane.ashiru-oredope@phe.gov.uk (D.A.-O.); ayoub.saei@phe.gov.uk (A.S.); rosie.allison@phe.gov.uk (R.A.); cliodna.mcnulty@phe.gov.uk (C.A.M.M.); 2School of Pharmacy, University of Nottingham, Nottingham NG7 2RD, UK; tracey.thornley@boots.co.uk; 3Boots UK, Nottingham NG90 1AA, UK; aimicharlottedickinson@hotmail.co.uk (A.D.); clare.x.ingram@boots.co.uk (C.I.); 4School of Healthcare, University of Leeds, Leeds LS2 9DA, UK; philip.howard2@nhs.net; 5NHS England, Leeds LS2 7UE, UK

**Keywords:** mixed-method, qualitative, questionnaire, infection self-care, antimicrobial resistance, antimicrobial use, e-learning, behavioural science

## Abstract

The community pharmacy antimicrobial stewardship intervention (PAMSI) is multi-faceted and underpinned by behavioural science, consisting of the TARGET Antibiotic Checklist, staff e-Learning, and patient-facing materials. This mixed-method study evaluated the effect of PAMSI on community pharmacy staffs’ self-reported antimicrobial stewardship (AMS) behaviours. Data collection included staff pre- and post-intervention questionnaires, qualitative interviews, and TARGET Antibiotic Checklists. Quantitative data were analysed by a multivariate ordinal linear mixed effect model; qualitative data were analysed thematically. A total of 101 staff participated from 66 pharmacies, and six completed semi-structured interviews. The statistical model indicated very strong evidence (*p* < 0.001) that post-intervention, staff increased their antibiotic appropriateness checks and patient advice, covering antibiotic adherence, antibiotic resistance, infection self-care, and safety-netting. Staff reported feeling empowered to query antibiotic appropriateness with prescribing clinicians. The TARGET Antibiotic Checklist was completed with 2043 patients. Topics patients identified as requiring advice from the pharmacy team included symptom duration, alcohol and food consumption guidance, antibiotic side-effects, and returning unused antibiotics to pharmacies. Pharmacy staff acknowledged the need for improved communication across the primary care pathway to optimise antimicrobial use, and PAMSI has potential to support this ambition if implemented nationally. To support patients not attending a pharmacy in person, an online information tool will be developed.

## 1. Introduction

Antimicrobial resistance (AMR) is exacerbated by the use of antimicrobials [1]. The UK’s 2019–2024 national action plan to tackle AMR aims to reduce unintentional exposure to, and optimise use of antimicrobials [2]. In England, the majority of antibiotics are prescribed in the community [3,4] and dispensed by community pharmacy staff who act as the last health care professional defence in safeguarding the appropriate use of antibiotics. An estimated 1.6 million people visit a pharmacy in England daily [5], making them some of the most used healthcare professionals [6]. The Community Pharmacy Contractual Framework 2019–2024 recognises the integral role of community pharmacy staff in prevention of illness in the community [7], and the Pharmacy Quality Scheme (PQS) provides incentives to increase their clinical and public health activities; AMS (Antimicrobial Stewardship) has been included as a PQS domain since 2020 [8].

There is evidence of antibiotic non-adherence behaviours of patients in the community, such as not completing the prescribed course, and using leftover or non-prescribed antibiotics [9,10,11,12]. The pharmacists and other team member’s roles in checking the appropriateness of antibiotic scripts and educating patients on appropriate antibiotic use may help improve antibiotic adherence and avoid unnecessary exposure to antimicrobials, thereby helping to reduce rates of AMR. The public trusts the advice of pharmacists on self-care and symptomatic treatment [13]. AMR has been described as a hidden pandemic; experts recognize the need to engage the public in appropriate antibiotic use and support continued heightened attention to infection prevention as displayed during the COVID-19 pandemic [4,14].

Community pharmacy staff believe that providing antibiotic and infection self-care advice is a key responsibility; however, lack of time, resources, and awareness of their role in AMS often prevents this [15,16,17]. The implications are that pharmacy teams need dedicated training and appropriately disseminated supporting tools for staff and patients, which fit into the dispensing journey [18]. The TARGET (treat antibiotics responsibly; guidance, education, and tools) Antibiotic Toolkit [19] provides AMS resources for primary care and includes a community Pharmacy Antimicrobial Stewardship Intervention (PAMSI). PAMSI was co-developed with pharmacists, other pharmacy staff, patients, designers, and researchers and was found to be a feasible intervention when trialled with staff from 12 community pharmacies and 1000 pharmacy patients [20]. Designed using the behaviour change wheel [21], PAMSI consists of staff AMS e-Learning, patient-facing materials, and a TARGET Antibiotic Checklist, which is completed by the patient (information on indication and knowledge of antibiotic use) and pharmacy staff (safety and appropriateness checks). The intervention aims to educate and empower staff in their AMS role, provide a set of cues for checking the antibiotic, and prompt staff to tailor advice to patients. Following on from the feasibility study [20], this effectiveness study aimed to evaluate staff AMS behaviours before and after using PAMSI, including the checking of antibiotic appropriateness and providing patients with tailored infection and antibiotic use advice based on identified need. Secondary aims were to identify facilitators and barriers to intervention implementation.

## 2. Materials and Methods

### 2.1. Study Design

The effectiveness of PAMSI was measured pragmatically in everyday community pharmacy practice. Quantitative and qualitative data were collected via pre- and post-intervention questionnaires, semi-structured interviews with community pharmacy staff, and a follow-up questionnaire with participating patients (Figure 1). Participating staff were invited to complete the AMS for Community Pharmacy e-Learning [22] and use the TARGET Antibiotic Checklist [19], posters, and patient-facing information leaflets (including TARGET ‘Treating Your Infection’ leaflets, an NHS dental leaflet, and an Antibiotic Research UK leaflet [23]).

### 2.2. Pilot Study

A pilot, using all questionnaires and interview schedules planned for the main evaluation, was completed with six purposively recruited community pharmacies (varied size and type) in South-West England in August 2020, to identify if the study design was still appropriate during COVID-19 restrictions. Staff implemented the PAMSI for two weeks and participated in a semi-structured telephone interview. The pilot supported the commencement of the full evaluation if the antibiotic prescription was being collected in person at the pharmacy, and if staff utilised local COVID-19 guideline measures or supported patients in completing the TARGET Antibiotic Checklist.

### 2.3. Setting and Participants

A total of 105 Boots UK community pharmacies located in England were purposively selected to include different regions, areas (e.g., town, city, village), and type of pharmacy (shopping centre, healthcare, train station). The number and range of pharmacies invited were based on practicality, available resources, and the goal to maximise diversity in participants. Pharmacies were invited via a centrally disseminated communication from Boots UK, followed by email correspondence from researchers. One staff member from each pharmacy provided informed consent and acted as the pharmacy representative; they were asked to return completed TARGET Antibiotic Checklists and encourage colleague involvement. Participating staff were asked to complete the AMS e-Learning and then use the resources for four to six weeks from October to December 2020.

### 2.4. Data Collection

(a)TARGET Antibiotic Checklists

In October 2020, pharmacies received a package comprising printed TARGET Antibiotic Checklists, patient-facing leaflets and resources (available via the TARGET toolkit [19]), and pre-addressed envelopes for the purpose of returning completed Checklists to an external agency; staff representatives were reminded weekly by email to return these. The agency inputted and returned data to the researchers on a weekly basis through a secure online platform; a portion of the data (20%) was double entered by the external agency and checked for accuracy by researchers; no issues were identified.

(b)Staff questionnaires

The pre (October 2020) and post (December–January 2021) staff intervention questionnaires (Appendix A) were hosted online and emailed to staff representatives and pharmacy managers to distribute to colleagues. Questions included a single option Likert scale on self-reported AMS practices. The primary outcome was reported checking of antibiotic appropriateness (antibiotic choice, dose, and course length). Secondary outcomes were reported provision of patient information on antibiotic adherence, antibiotic resistance, self-care, and safety netting, as well as reported queries to antibiotic prescribers.

(c)Staff interviews

Pharmacy staff were able to express interest in participating in interviews at the end of the post-intervention questionnaire; 16 of the 81 staff (20%) expressed interest and were invited in February 2021; six pharmacists agreed. Interviews were conducted by two female researchers (AT and FP) experienced in qualitative research who encouraged pharmacists to speak openly about their experiences. These researchers were not involved in the first part of the evaluation and therefore were not known to participants. The interview schedule (Appendix A) was aligned to the COM-B model [21] and explored experiences and barriers to implementing the intervention. Due to the 2020 COVID-19 face-to-face restrictions, interviews were completed by skype, Microsoft Teams, or telephone, according to participant preference, and where possible participants were encouraged to have their video turned on. Researchers made field notes and discussed emerging themes to probe in subsequent interviews. Pharmacy staff who participated in interviews did so in their own time and were provided with a £25 honorarium. Discussions lasted between 30 and 45 min and were recorded, transcribed verbatim by an external agency, and checked for accuracy by researchers.

(d)Patient questionnaire

In addition to the main data collected, a questionnaire was sent to patients who provided their contact details (email address or telephone) on the TARGET Antibiotic Checklist. Checklist data were returned on a weekly basis, and researchers followed up all patients who provided contact details, with a link to an online questionnaire. As this was not a primary outcome, the results are provided in the Appendix A.

### 2.5. Data Analysis

Microsoft Excel was used for descriptive analysis and visual presentation of the quantitative data. A statistical model was applied to staff pre- and post-intervention questionnaire primary and secondary outcomes. The aim of the multivariate ordinal linear mixed effect modelling was to detect evidence of improvements in the primary (staff checking of antibiotic appropriateness) and secondary outcomes (provision of advice to patients) at post- versus pre-intervention. The linear predictor included individual random (latent) effects to consider extra variation that was not explained by the intervention. The linear predictor included additional variables collected from participants in addition to the intervention: participant professional role; pharmacy region; rural or urban area; whether the pharmacy had taken part in previous infection-related interventions; and participants’ previous training in infections or antibiotics. All statistical analysis was completed in the generalised latent linear and mixed model (GLLAMM) in Stata 17.

Analysis of qualitative interview transcripts followed a six-stage inductive thematic analysis consisting of: familiarisation with the data; coding the meaning of data in the transcripts; searching for themes; naming, reviewing, and revising themes; and reporting themes [24]. NVivo pro-11 was used to organise thematic analysis. AT analysed all interviews, and FP analysed 25%; throughout analysis, researchers regularly discussed any differences in coding, their own beliefs, and insights on the data. Themes were discussed and agreed upon by the research team. Quotes which illustrated the themes were identified for reporting.

### 2.6. Ethics

This service evaluation was internally reviewed and approved by the UKHSA Research Ethics and Governance Group (REGG) (Reference: R&D NR0176). All participants providing data via questionnaires or interview were provided with information and consent forms before consenting to participate and were aware of the study aims. Questionnaires were hosted on a secure SnapSurvey platform, and all files were handled in accordance with the Data Protection Act 2018 and General Data Protection Regulations (GDPR).

## 3. Results

A total of 101 pharmacy staff from 66 of 105 (62.9%) invited community pharmacies participated; 91 and 81 pharmacy staff completed the pre- and post-intervention questionnaires, respectively, and eight pharmacies were lost to follow up. Most staff identified as a pharmacist (59%) or dispenser (18%); see Table 1 for participant characteristics.

Pharmacy staff used the TARGET Antibiotic Checklists with 2043 patients over the data collection period; 884 patients provided contact details, and 89 (10%) responded to a follow up questionnaire after their pharmacy visit (Appendix A).

### 3.1. TARGET Antibiotic Checklist Results

Of the TARGET Antibiotic Checklists completed by participating pharmacy staff, 1944/2043 (95%) of patients named the infection they were being treated for, and 1846/2043 (90%) patients reflected on their knowledge upon arrival at the community pharmacy (Figure 2). While patient knowledge was high, areas requiring advice from the pharmacy team were symptom duration (20%), antibiotic side effects (13%), food consumption (13%), alcohol consumption (8%), and returning unused antibiotics to pharmacies (8%); 71% of pharmacy staff reported using this knowledge assessment to guide the advice they provided to patients, and they indicated that an infection self-care leaflet was given to 1276 patients (62%).

Pharmacy staff also used the TARGET Antibiotic Checklist to audit antibiotic safety and appropriateness checks against each prescription. Staff indicated that they checked for: allergies and risk factors (84%); the correct dosage (86%); the correct duration (84%); antibiotic appropriateness (86%); and that they checked against local prescribing guidance (73%). The most common antibiotics dispensed were amoxicillin (31%), nitrofurantoin (16%), and flucloxacillin (15%).

### 3.2. Pharmacy Staff Questionnaire Findings

Antibiotic Appropriateness

Most staff members (81/91, 89%) completing the pre-intervention questionnaire reported that checking appropriateness of antibiotic prescriptions was part of their role, and 41% reported that they always or very often checked antibiotic appropriateness (Figure 3); this increased to 79% (61/81) post-intervention. Statistical modelling suggested strong evidence (*p* < 0.001) of increased antibiotic appropriateness checks. Post-intervention, fewer staff reported ‘never’ querying an antibiotic prescription with a prescribing clinician; however, statistical modelling found no significant evidence of this.

#### 3.2.2. Providing Advice

A total of 90/91 staff responding to the pre-intervention questionnaire and 70/81 staff responding to the post-intervention questionnaire indicated they had a role in providing advice to patients. The percentage of staff self-reporting how often they provided advice to patients increased across the seven topics (Figure 4). The provision of advice on the topic of antibiotic resistance was reported the least by pharmacy staff in the pre-intervention questionnaire, followed by duration of symptoms and signs of more serious illness. Statistical modelling suggested strong evidence (*p* ≤ 0.01) for an increase in provision of advice across six of the seven topics, at post-intervention, except for ‘how and when to take the antibiotic’, likely due to the high response at pre-intervention.

Staff were asked about communication methods for providing infection and antibiotic advice (Figure 5), which included nine options relating to advice provided verbally (in person or phone), through written and hard copy leaflets, posters, and signposting to apps or websites. Most staff reported providing advice verbally (35% always) in the pre-intervention questionnaire; very few reported other methods. Statistical modelling found 7/9 of the communication methods improved post-intervention, particularly providing patients with hard copy leaflets, directing them to a screen or poster, and signposting to a website via text message (*p* < 0.001).

Post-intervention, staff estimated that the extra time to provide tailored advice to patients with the TARGET Antibiotic Checklist was an average of 5 min per patient. Most staff agreed that this extra time was feasible (27% strongly agree, 44% agree) and justified by the benefits of keeping antibiotics working (49% strongly agree, 36% agree).

### 3.3. Pharmacy Staff Interview Themes

Six community pharmacists participated in semi-structured interviews to share their experience of implementing PAMSI. Three of the six had additional roles as pharmacy managers. See Table 2 for a summary of the themes.

#### 3.3.1. Theme 1: Enablers for Embedding into Practice

Staff reported the TARGET Antibiotic Checklist and patient information leaflets fitted into their working day, and the extra time required was feasible as it was necessary information. Incorporation into routine was facilitated by embedding the materials in the everyday environment (placing leaflets on the counter) and involving the whole pharmacy team (encouraging all staff to complete the e-Learning). Staff felt that PAMSI fitted into wider priorities, including the 2020/2021 PQS, and that it improved team motivation. The TARGET information leaflets were useful for patients attending the pharmacy with minor ailments without an antibiotic prescription, particularly urinary tract infections (UTIs). Staff reported that the e-Learning improved their knowledge of AMR and AMS and helped their team to understand the importance of their role in reducing AMR.

#### 3.3.2. Theme 2: Perceived Benefits

Overall, staff felt their team were more confident in querying prescriptions with a prescriber when necessary, and that the TARGET Antibiotic Checklist was useful to engage the prescriber in conversation. Staff believed communication with patients had improved and they felt more aware of areas where patients lacked knowledge, which helped them initiate conversations that would not have occurred otherwise. All interviewees said they would continue to use PAMSI, as they felt they increased patient engagement and improved the safety and effectiveness of their medicine.

#### 3.3.3. Theme 3: Barriers to Intervention Implementation

COVID-19 affected implementation of the PAMSI tool through changes in general practice (GP) and dental prescribing and fluctuating patient attendance throughout the pandemic. Some staff felt there was a lack of AMS continuity across the patient pathway, and that more shared working was needed between pharmacies, GPs, and dentists. In some cases, it was necessary for staff to complete the patient checklist sections on behalf of the patients by asking questions verbally. Some staff felt this was more effective, as it was helpful to explain the purpose of the questions, while others felt patients might not be honest about where they lacked knowledge if this verbal approach was used. There were also concerns about some patients who did not fully complete the TARGET Antibiotic Checklist. Staff reported a lack of time to use the resources for every patient collecting an antibiotic, and many prescriptions were collected by representatives or delivered directly to the home. Although information was given to the representative or leaflets were included in the delivery, staff were unsure if patients had engaged with the information as there was lack of follow up. A minority of staff reported language barriers but was able to overcome these through members of staff who could speak the language; however, translated resources would be useful.

## 4. Discussion

PAMSI supported community pharmacy staff to significantly improve AMS behaviours around assessing appropriateness of antibiotics and providing antibiotic and infection advice to patients. This was facilitated by whole team involvement and embedding the intervention into routine. Staff believed PAMSI supported essential roles, and therefore the extra time to provide tailored advice to patients was feasible and justified. Qualitative themes were that staff felt motivated in their AMS role through improved knowledge of AMR, felt more capable to query decisions by other prescribers, and had greater opportunity to provide tailored information through understanding patient knowledge gaps. Supplementary findings from follow up with 89 patients suggest positive impacts, as the majority was able to recall the advice provided to them at the pharmacy and reported good knowledge of their antibiotics; however, further research is needed to ascertain impacts of PAMSI on patients.

Similarly to the feasibility study of PAMSI, staff reported that the intervention fitted into their operational practice and led to more meaningful conversations [20]. Our findings go further by demonstrating that embedding PAMSI can significantly improve AMS activities of pharmacy staff. Historically there have been barriers for community pharmacy AMS due to lack of access to patients’ notes and lack of connections between pharmacies and other care settings [15,16,17]. The TARGET Antibiotic Checklist component of PAMSI was designed to account for this by collecting information from the patient that the pharmacy staff require to assess the prescribed antibiotic. The e-Learning component was designed to improve pharmacy staffs’ capability and motivation in their AMS role, and in our qualitative interviews, staff highlighted that the e-Learning helped the team to understand the importance of their AMS role. The pharmacy staffs’ AMS role can reduce unnecessary exposure to antimicrobials and ensure better outcomes for patients; however, existing research suggests this role is not fully embedded. A 2016 questionnaire study found that 12% of 50 pharmacists reported checking prescriptions against local prescribing guidelines, and while staff often gave advice on dose and completing the course of antibiotics, they rarely gave infection self-care advice [25]. The improvement in the provision of advice we found may be because pharmacy staff can gather insights on where their patient population lacks knowledge with the TARGET Antibiotic Checklist and can share specific patient information leaflets accordingly. Similarly, a randomised control trial (RCT) of an educational webinar for community pharmacists, followed by use of the TARGET Respiratory Tract Infection leaflet, increased self-care advice given and reduced referrals to GPs [26]. Our intervention builds on this by embedding the TARGET Antibiotic Checklist into the dispensing journey, so that AMS learning can be implemented on an ongoing basis and become routine.

Although the TARGET Antibiotic Checklist can act as an audit to ensure the relevant checks have been carried out on all prescriptions, we found that these were not done for 15–25% of the 2043 Antibiotic Checklists completed with patients, suggesting environmental barriers preventing implementation. Interview participants in our study commented on the impact of time pressures and patients not being physically present (heightened by the COVID-19 pandemic), as well as lack of continuity along the patient pathway. Access to local antibiotic prescribing guidelines may be a factor, and therefore commissioners should ensure these are disseminated locally and easily available. The lack of easy communication between pharmacy and general practices has been previously highlighted, and staff have suggested the need for better referral pathways [27]. When trialled in community pharmacies, the TARGET Urinary Tract Infection (UTI) patient information leaflet was suggested by pharmacy staff as a method to provide information to other healthcare professionals, for example when referring patients with UTI symptoms to GP surgeries [28]. Previous qualitative research suggests that pharmacy staff lack confidence to query other healthcare professionals’ prescribing decisions [15,16,27]. Although we did not find a significant increase in reported behaviour of pharmacy staff around querying antibiotics in the post-intervention questionnaire, in our interviews, staff reported feeling more empowered and motivated to question prescribing decisions when necessary.

Minor ailment schemes encourage patients to visit the pharmacy for common infections rather than general practice and could save £12 million per year if implemented across England [29]. The TARGET treating your infection leaflets, a component of PAMSI, could work alongside these schemes by supporting pharmacy staff providing infection self-care and safety netting advice to patients on common infections. Of 2043 TARGET Antibiotic Checklists completed, patients’ reported knowledge was similar to that found in the feasibility study of PAMSI [20]; the lack of understanding of duration of infection and returning unused antibiotics to the pharmacy is common in the literature [20,25,30]. A 2020 public survey on infection health-seeking behaviours found high trust for pharmacists’ advice about the need for antibiotics, which has significantly improved since a previous survey in 2014 [13]. The public trust the advice of community pharmacy staff on common infections [28,30]; however, public campaigns should continue to encourage this, as in public surveys, ‘visiting a GP’ is still the most common source of health information [31]. Use of online health advice is increasing, particularly in younger age groups [31]; however, our findings suggest that pharmacy staff primarily provided advice verbally, and very few signposted to websites or apps; although this did improve in the post-intervention questionnaire, this is likely due to COVID-19 pandemic restrictions. Since this evaluation, the TARGET patient information leaflets have been developed as accessible webpages which can be texted or emailed to patients [19], and these and other online information sources should be promoted as use of digital tools increases in pharmacy settings.

## 5. Strengths and Limitations

This mixed-method design informed by behavioural science allowed us to capture data on the effect of PAMSI on community pharmacy AMS, as well as understand behavioural insights into implementation. Collaborating with a central pharmacy chain allowed streamlined communication with pharmacy staff and helped to recruit a large sample; although we had some loss to follow up, this was limited considering the COVID-19 pandemic. A potential bias is that staff in other chains or independents may have experienced PAMSI differently; this effect was minimised by selecting a range of pharmacy types; however, other variables such as organisational culture may have influenced findings. Pharmacies were recruited centrally to avoid bias, i.e., recruiting pharmacies with an interest in AMS, and this is supported through the pre-intervention data showing low baseline AMS activity and low previous experience of interventions and training on AMS. As semi-structured interviews were an opportunity that participants could opt-in for, we were restricted to a smaller sample, and it is possible these participants may have had a biased interest in AMS; therefore, findings may not represent the views of all the staff who participated in the study; however, the interview themes are valuable as they support and give context to the quantitative findings.

Strengths and weaknesses of the development of PAMSI have been published elsewhere [20]; a possible weakness of the TARGET Antibiotic Checklist was that a large proportion of patients reported ‘yes’ to understanding all of the statements about their antibiotics, and there may be some acquiescence bias. The questions have since been updated to be more open and reflective. A limitation of the before and after study design was that within the given resources, outcome measures were self-reported by pharmacy staff, which may be subject to recall or acquiescence biases. Without access to the patient records it was difficult for pharmacists to support prescribers by validating the antibiotic prescription; however, the additional information collected in the TARGET Antibiotic Checklist allowed some insights. For the supplementary patient follow up questionnaire, there was low response rate and lack of diversity in the demographics, suggesting a biased sample. Future research should compare pharmacy users’ behaviours against a control group to measure intervention effects.

## 6. Implications and Recommendations

Since this study, the English PQS has included the AMS e-Learning and TARGET Antibiotic Checklist as criteria in 2020/21 and 2021/22, respectively. This will support embedding PAMSI into community pharmacies in England, and future work should evaluate the effect nationally. Guidance and training for staff and future policy considerations could promote the use of information leaflets and webpages, such as the TARGET toolkit [19], in facilitating discussions on the topics where patients reported lacking knowledge, such as self-care, preventing infections, and duration of symptoms. To overcome the barrier of patients not attending the pharmacy in person, an online TARGET Antibiotic Checklist tool is being developed. The TARGET Antibiotic Checklist and patient information leaflets [19] are available in multiple languages and should be issued to non-English speaking patients by pharmacy teams where appropriate.

Our study has highlighted that community pharmacy staff feel there is a lack of continuity and communication around AMS within the primary care pathway. There is a need for further research and policy considerations on how community pharmacy teams can be further integrated across the primary care pathway, including GP, dentistry, and out of hours.

## 7. Conclusions

The Pharmacy Antimicrobial Stewardship Intervention provides a model for community pharmacy AMS and facilitates tailoring information on antibiotics and common infections to patients’ needs. Pharmacy staff acknowledge a need for improved communication and integration across the primary care pathway to optimise prescribing and use of antimicrobials.

## Figures and Tables

**Figure 1 healthcare-10-01288-f001:**
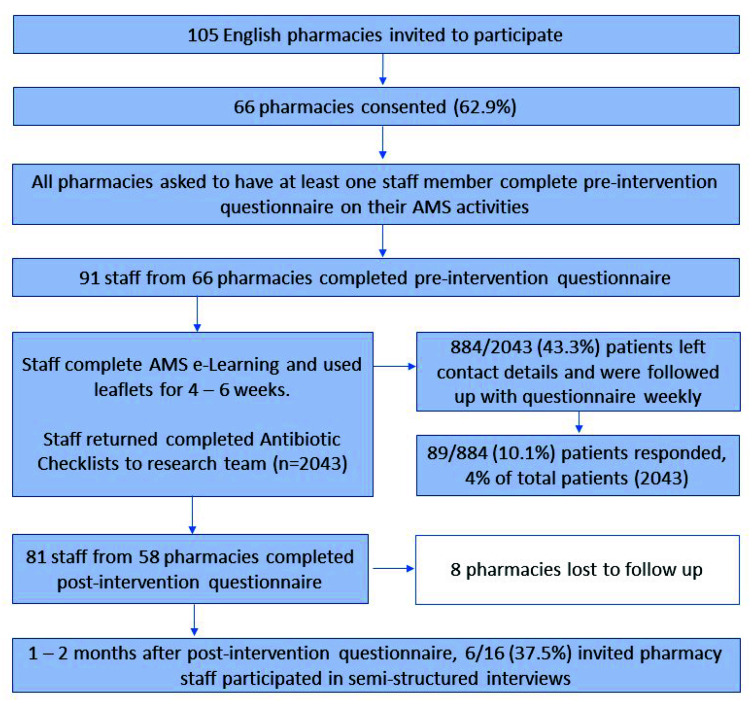
Mixed-method data collection process showing the study progression, methods, and participants.

**Figure 2 healthcare-10-01288-f002:**
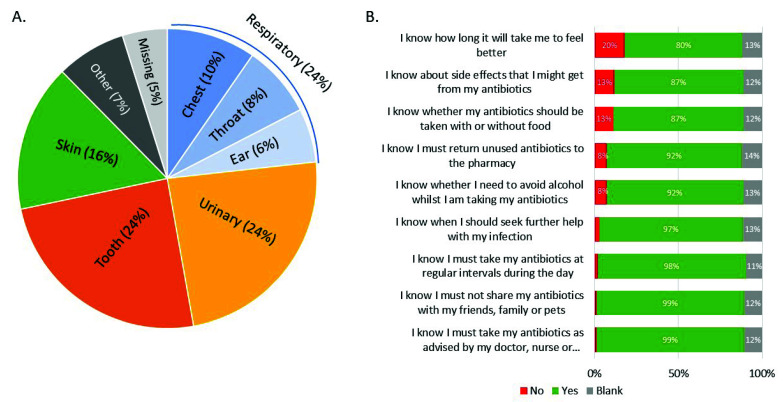
Patients’ (*n* = 2043) infection (**A**) and self-reported knowledge of antibiotic and infection management (**B**). Patients could select ‘yes’ or ‘no’ to understanding each of the nine statements.

**Figure 3 healthcare-10-01288-f003:**
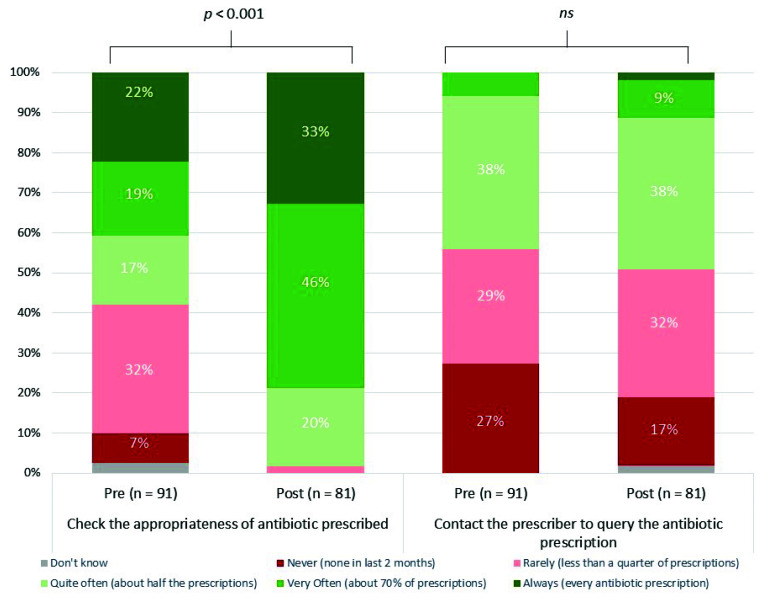
Community pharmacy staff reported checking of antibiotic appropriateness and contacting the prescriber to query the prescription, pre- and post-intervention. P signifies the strength of evidence from the statistical modelling for an improvement in the reported behaviour at post-intervention. Ns signifies there is no evidence of a significant improvement in behaviour at post-intervention.

**Figure 4 healthcare-10-01288-f004:**
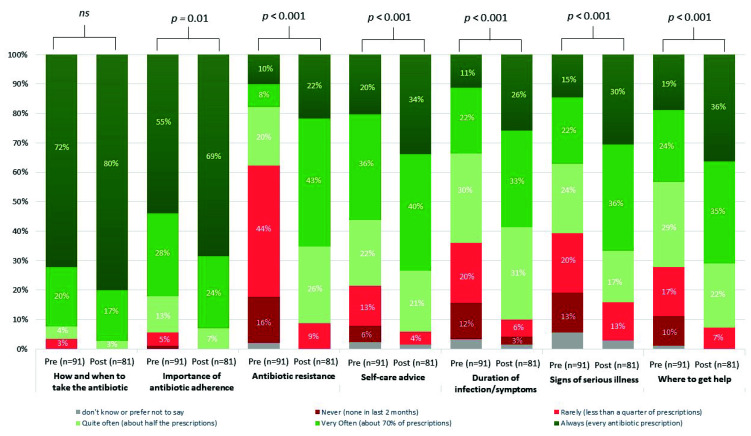
Community pharmacy staff provision of antibiotic and infection advice to patients, pre- and post-intervention. Pharmacy staff reported how often they provided advice to patients on their antibiotics and managing their infection. *p* signifies the strength of evidence from the statistical modelling, Ns signifies there is no significant evidence.

**Figure 5 healthcare-10-01288-f005:**
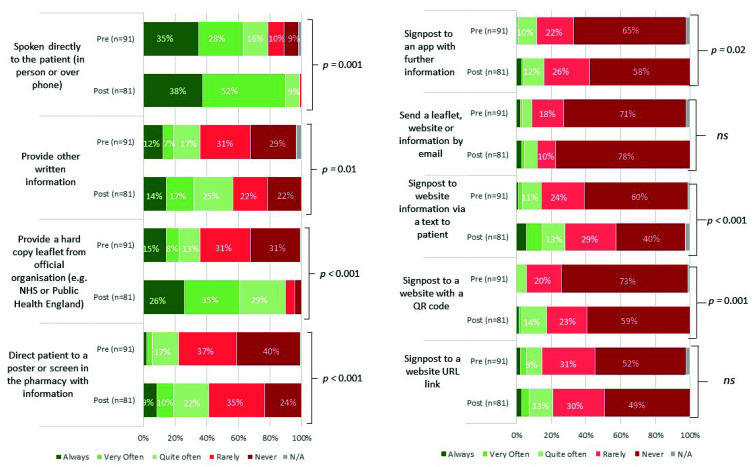
Community pharmacy staff method of advice to patients, pre- and post-intervention. Pharmacy staff reported what methods they utilised to provide advice to patients on their antibiotics and managing their infection. *p* signifies the strength of evidence from the statistical modelling, Ns signifies there is no significant evidence.

**Table 1 healthcare-10-01288-t001:** Self-reported demographic and role-related data of community pharmacy staff who participated in questionnaires.

Pharmacy Staff Characteristics (*n* = 101)	Number (%)
Role	
Pharmacist	60 (59%)
Dispensary team/dispenser	18 (18%)
Pharmacy technician	7 (7%)
Pharmacy manager	11 (11%)
Pre-registration pharmacist	4 (4%)
Trainee technician	1 (1%)
Has participated in previous AMS/IPC intervention	
Yes	10 (10%)
No	75 (74%)
Do not know or missing	16 (16%)
Has had previous training on AMS or IPC	
Yes	21 (21%)
No	54 (53%)
Do not know	17 (17%)
Missing	9 (9%)
Type of pharmacy they work in	
Town centre/high street	31 (30%)
Edge of town	7 (7%)
City Centre	13 (13%)
Small London store	7 (7%)
Health centre	5 (5%)
Retail park/shopping centre	8 (8%)
Train station pharmacy	5 (5%)
Rural	14 (14%)
Rural (small pharmacy)	11 (11%)
Region of pharmacy they work in	
South West	29 (28%)
East of England	14 (14%)
Central England	9 (9%)
London	20 (20%)
West Midlands	4 (4%)
South East	21 (21%)
North West	4 (4%)

AMS—Antimicrobial Stewardship. IPC—Infection, Prevention, and Control.

**Table 2 healthcare-10-01288-t002:** Staff themes and subthemes on facilitators and barriers to implementing the PAMSI. Mapped to Capability (C), Opportunity (O), Motivation (M) model.

Theme	Subtheme	COM	Quote
Enablers for embedding into practice	Extra time to implement resources is feasible and justified in everyday practice	O	*“Yeah I think it should be part of your normal practice. If someone’s got something that needs explaining it needs explaining.”* *“…because some people say, no, that’s miles too much time. But, again, if care is at the forefront then what’s five minutes?”*
	Use of Antibiotic Checklist and leaflets became part of routine	O	*“…when I’m doing a consultation and I give the leaflet out to people then, just because it’s part of our consultation routine already”* *“We have the wallet system on the counter so we put it in with the prescription in the wallet so that when the girls got the prescription out they could see that it needed to be part of the survey.”*
	Whole team involvement to implement the resources	O M	*“so it wasn’t just the pharmacists doing it, we had the whole team on it, doing the checklist and handing out, at the handing out process, because part of handing out process they do, do that… So they were confident doing the checklist.“*
	Leaflets and checklist worked in conjunction to facilitate conversations	O	*“…the number of patients we saw that were coming in with minor infections and ailments…we could use the leaflets at that stage to talk to patients”* *“I think because we were having more of an active conversation about the antibiotics, and then you can make reference to the leaflet and it ties it all in together for the patient”*
	Implementation fit with wider priorities	O M	*“It* [AMS e-Learning] *was part of our Pharmacy Quality Payments Scheme, there’s a domain that we have to fulfil around antimicrobial resistance.”*
	e-Learning helped understand justification	M	*“So, I think, more than anything, the e-Learning was what made me evaluate my practice and try and promote antibiotic resistance as much, more than I did at the time.”*
Perceived benefits	Staff felt confident and encouraged to query prescriptions	C O M	*“But the reverse part which checklist brought back to me was that like, do you actually need this* [antibiotic] *at this point, is what was very interesting.”* *“So from a team point of view, materials helped give them confidence that they had something to use as a resource to have the conversation* [with a prescriber].”
	Conversations with patients were more effective	C O M	*“…that’s a conversation that wouldn’t have ever occurred had we not had the checklist.”* *“I think* [completing antibiotic checklist] *was really helpful actually. I think it draws the patient’s attention more to what they might not realise.”*
	Beliefs about benefits to patients	M	*“I was answering questions that I would not usually be asked by patients, so I think they were gaining a lot from the checklist, a lot more knowledge about their medications.”* *“…they’re* [the patients] *using their antibiotics more effectively and getting more optimal effects from them, and that’s what we want our patients to get at the end of the day”*
Barriers to intervention due to COVID-19 and other contextual factors	Change in prescribing habits and fluctuating patients	O	*“No, we just, what had happened during COVID-19 is the dentists weren’t really seeing patients, so they were emailing us prescriptions for antibiotics.”* *“…there was definitely a reduction of the amount of checklists that we would have completed because we had a higher volume of patients coming in before the pandemic. And so, we would have had more prescriptions and people completing the checklist.”*
	Importance of continuity along patient pathway	O	*“…maybe some highlighting or awareness to GP surgeries as well, but this is what they’re doing. And maybe that would get them to reflect on their practice too because I think, as much as we change our practice, it’s also really important for GPs and prescribers to have that awareness and that extra thought that we now have, is this really necessary or is it the correct duration, and things like that.”* *“…but I do feel that there’s more conversations to be had between the GPs and the pharmacists of the nominated pharmacy that it’s going to.”*
	Patients did not want to interact with physical materials	O	*“In terms of any other barriers, I would say the main barrier that I faced was a lot of people didn’t want to touch it* [the Antibiotic Checklist] *after I had touched it.”* *“There wasn’t really anything preventing them from giving the leaflets but, we had to be mindful of infection control when using pens and passing them back and forth.”*
	Time barriers	O	*“We couldn’t just ask them to sit down and do it. So yeah, it was a little bit more time consuming than I suppose it would have been in normal times but quite helpful.”*
	Patients not physically present	O	*“…sometimes I think you’re wanting to make sure the message landed once the prescription got back to the patient and there was no way of checking that.”* *“Maybe not so many for the delivery patients because it’s difficult to have a conversation with them and they’re not quite sure why they’re being given medication anyway.”* *“When a representative came in to collect a prescription on behalf of the patient, I would add to the conversation I had put some leaflets inside, please get* [the patient] *to give me a call if they have any questions...And I didn’t get anything.”*
	Belief that not all patients engaged with checklist	O	*“So I personally think and because it’s such a long questionnaire as well handing it over the patient, I don’t think we would get a realistic response, but I would definitely try it.”* *“…when you hand something over to patient to complete, I find, this is my personal experience, that they don’t read all the questions, they just skim read it and tick it off”* *“sometimes it makes it easier for the patients to understand and for you to tick for them.”*
	Language barriers	O	*“…but then again if you have a pharmacist who can speak that language it was well signposted to the pharmacist directly in the first place. So they saw a barrier, they overcome the barrier, but I wouldn’t say they overcame that 100% of the time.”*

## Data Availability

The corresponding author can be contacted for further data and study materials upon reasonable request.

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
