# Peer review of "Mixed-Method Evaluation of a Community Pharmacy Antimicrobial Stewardship Intervention (PAMSI)"

_healthcare, 2022, doi:10.3390/healthcare10071288_

Round 1
Reviewer 1 Report
Thank you for the opportunity to review this study on the evaluation of a community pharmacy antimicrobial stewardship intervention.
The main objective of this study was to demonstrate the effect of an antimicrobial stewardship intervention on pharmacy staff AMS behaviours. This objective suggests a feasibility study when it is not clearly stated. A primary efficacy endpoint (e.g. appropriateness of the antibiotic) and another design (cluster randomized trial, stepped-wedge design) would have been more appropriate for a large study like this involving so many patients and healthcare professionals.
Moreover, the before-and-after methodology increases the risk of bias. Furthermore, the results are too numerous and not prioritized, resulting in a confusing message. This is a feasibility study presented as an efficacy study.
Author Response
We would like to thank the reviewer for their comments. Regarding the comments about the study design, we respectfully disagree that this is a feasibility study, but nor is it an efficacy study and we are grateful to the reviewer for highlighting that this was not clear. As stated in the manuscript this study followed on from a feasibility study, which found that the intervention was feasible, practical, and acceptable to community pharmacy staff. The current study had the objective of investigating the effect of PAMSI on pharmacy staffs' antimicrobial stewardship behaviours. This is an effectiveness study, and as such we took a pragmatic approach and aimed to investigate the effect of the intervention in everyday community pharmacy practice, which would not be possible in a highly controlled efficacy study context. We have clarified that this is an effectiveness study in the manuscript, and hope this is clearer to reviewers.
We agree that there is risk of bias with our study and have discussed this within the existing limitations sections.
We agree that there are numerous results in this manuscript as we originally presented both data from pharmacy staff and patients. As the main objective was to investigate community pharmacy staff behaviours, we have amended the results section to clearly focus on pharmacy staff, and moved the patient follow up survey data to the supplementary as this was not a primary outcome. We have also provided a clear statement of findings at the beginning of the discussion and subheadings throughout to help present the different results. We hope this helps provide a clearer story to the manuscript.
Reviewer 2 Report
This was an excellent report on very thorough research. I have a few substantive questions/comments to consider:
- The authors said in the abstract that the statistical model indicated very strong evi-22 dence (p<0.001) that post-intervention. However, there is nothing mentioned about this point in the statistical analysis method.
-There was no mention of the limitations of the study. Also, mention how your results compare to another study, which was published very recently.
Author Response
We thank the reviewer for their kind comments on the manuscript. Regarding the comments/questions from the reviewer, we believe these were covered in the original manuscript and therefore have not made any amendments. We would like to direct the reviewer to the results section ‘3.2. Pre- and Post-intervention questionnaire findings’ where the results of the statistical model are presented. We would direct the reviewer to the discussion section ‘5. Strengths and limitations’ which has substantial discussion of limitations. It is not clear which other study, published very recently the reviewer is referring to; it may be the feasibility trial of our intervention which we have referenced in the introduction and discussion (reference: Allison, R., Chapman, S., Howard, P., Thornley, T., Ashiru-Oredope, D., Walker, S., Jones, L.F. and McNulty, C.A. Community pharmacy campaign to Keep Antibiotics Working (KAW): An innovative new approach to improve patients’ understanding of their antibiotics. JAC-Antimicrobial Resistance 2020, 2, p.dlaa089, doi:10.1177/1757177419868911.).
Reviewer 3 Report
The comments as below.
1. The keywords should be specific for the manuscript. Keywords such as “before and after” and “implementation” should be removed.
2. Why only 105 pharmacies are invited? How did you calculate the sample size?
3. The patients’ self-reported knowledge (figure-2), the responses have been taken in two different formats where the questions (before advice) are different than the after advice. Further, the response was taken using a 6 Likert scale after advice. The reflection of knowledge and its measurement is different (Pre-advice vs. post-advice), how did you compare the response? A vis-a-vis comparison should have been done.
4. Is there any data captured for the antibiotics that are most frequently used by the patients. Probably the researchers could have given the information on the antibiotic categories like UTI, RTI, or others. I understand this information may not have much influence on the knowledge of patients or pharmacists however, the duration of medication use and types of infection might influence the results.
Overall, the research is interesting and needs a minor revision.
Author Response
We thank the reviewer for their comments to the manuscript and hope the revisions make these areas clearer.
We have removed the suggested key words and thank the reviewer for highlighting this.
Regarding the sample size, as this was not an efficacy or controlled trial we did not do a sample size calculation, however we based the number of community pharmacies to invite on practicality and available resources while allowing a range of diversity in terms of type and locations of pharmacies.
We agree that the patients self-reported knowledge (as shown in the original figure 2) cannot be directly compared as these were asked differently at post-intervention. As the main objective of the study was the effect of the intervention on pharmacy staffs’ antimicrobial stewardship, it was not a primary aim to investigate improvements in patient’s knowledge. Therefore, we have amended the results section to make clearer the focus on pharmacy staff and have included the patient follow up survey data in the supplementary as additional information for readers. This was supported by comments from another reviewer.
Thank you for suggesting giving more information about the infections and antibiotics used most frequently by patients, we agree this would be interesting to include and have amended figure 2 to include information on the patients infection.